# Qualitative study exploring barriers and facilitators to progression for female medical clinical academics: interviews with female associate professors and professors

Diane Trusson ![ORCID] , Emma Rowley

School of Medicine, NIHR Applied Research Collaboration East Midlands (ARC EM), University of Nottingham, Nottingham, UK

**Correspondence to**
Dr Diane Trusson;
diane.trusson@nottingham.ac.uk

## ABSTRACT

**Objectives** This study aimed to explore the barriers and facilitators to career progression for female medical clinical academics from the perspectives of female associate professors and professors, with a particular focus on women with caring responsibilities.

**Design** An exploratory qualitative approach was adopted. Data from semistructured interviews conducted via video calls were analysed using thematic analysis.

**Setting** Two major universities in the East Midlands of England.

**Participants** The sample consisted of 13 female medical clinical academic associate professors and professors representing a range of medical specialties.

**Results** Female medical clinical academics experienced barriers and facilitators to progress at individual, interpersonal, institutional/procedural and societal levels.

**Conclusions** Many barriers experienced at an individual level by female medical clinical academics are heavily influenced by their interpersonal relationships, the academic environment in which they work and broader institutional and procedural issues which, in turn, are influenced by stereotypical societal views on gender roles. Facilitating factors, including measures to increase the numbers of female leaders, may lead to a change of culture that is supportive to aspiring female clinical academics as well as enabling a healthy work/life balance for women and men with caring responsibilities.

## Strengths and limitations of this study

► Adopting a qualitative approach allowed for an exploration of medical clinical academic experiences from the participants' perspectives.
► There was a survivor bias in that the study only sought the views of women who had achieved associate professor or professor.
► The study did not include female associate professors/professors who are nurses, midwives or allied health professionals.
► The study was limited by a small sample from one geographical area.

## INTRODUCTION

Currently in the UK, fairly equal numbers of male and female medical clinical academics reach clinical lecturer level.[1] However, 'the percentage of female fellows declines with increasing seniority of award'[2] (p 7), with women representing just 22.1% of UK professors in 2020.[3]

The absence of female leaders and the associated costs to research and teaching is of concern both nationally and internationally.[4–6] Penny *et al* stress the importance of addressing gender imbalances to avoid biasing the research agenda (and consequently future clinical practice), wasting talent and negatively affecting the culture in the clinical academic research system.[7]

A 2001 study identified the barriers to progression for female medical clinical academics at individual and institutional levels,[8] yet 20 years on, it seems that 'little progress has been made in retaining female clinical academics and facilitating their progression to senior leadership roles…and barriers remain to be addressed'[3] (p 25).

Difficulties in balancing clinical and academic roles along with family responsibilities are well documented, with research suggesting that female medical clinical academics' careers are more likely to be adversely affected by shouldering more domestic responsibilities than their male colleagues.[5 6] Therefore, it is important to identify ways of better supporting women combining a clinical academic career with family life, and to 'create an environment that accepts the realities of a work-life balance'[4] (p 1056). It is also important to consider the 'broader structures in which academic institutions are situated' which influence the 'culture, policies and practices' in which

female medical clinical academics work[9] (p 503). For example, male-dominated institutional cultures 'heavily influenced by stereotypical beliefs and a lack of gender equality policy implementation'[10] (p 14) can present barriers to progression including a lack of female role models and mentors, and implicit bias in research and recruitment practices.[11]

The Athena SWAN charter was introduced in the UK in response to concerns surrounding gender disparities in higher education and research. It 'encourages and recognises commitment to advance women's careers in key areas including representation…career milestones, and working environment for all staff'[12] (p 2). The achievement of (at least) Silver Athena SWAN status has been a prerequisite for medical schools to receive funding from the National Institute for Health Research since 2011.[12] However, previous research indicates that simply having policies in place does not always translate into gender equity in women's lived experiences.[13]

The aim of this study was to explore barriers and facilitators to medical clinical academic career progression from the perspectives of female associate professors and professors.[11 13]

## METHODS
This manuscript has been prepared using the Consolidated Criteria for Reporting Qualitative Research checklist (see online supplemental file 1).

### Study design
A qualitative methodology was used as is the most appropriate way of exploring individual subjective experiences in depth.[14]

### Recruitment
School administrators distributed emails with details of the project to associate professors and professors based in the medical schools of two major universities in the UK. Female associate professors and professors who were willing to be interviewed were invited to contact the lead researcher. Snowball sampling enabled access to further potential participants. The sample of 13 represents a relatively high response rate given the low numbers of women in these roles and the timing of the research which coincided with the COVID-19 pandemic.

### Data collection
Interviews took place between October 2020 and February 2021. They were conducted by the first author via telephone or video calls due to COVID-19 restrictions. Online consent forms were obtained before interviews commenced. Semistructured interviews followed the same interview guide (see online supplemental file 2) but also allowed interesting points to be followed up with probes for further information.[14] Interviews were digitally recorded (with permission) and lasted between 25 and 60 min.

### Research team
The (all female) research team was led by the first author, a research fellow with a background in medical sociology and extensive experience of conducting social research. Other team members included a medical professor, an associate medical professor, a senior research advisor, an Athena SWAN operations manager and a capacity development manager.

### Data processing
Interviews were transcribed professionally with all identifying data removed prior to analysis.

### Analysis
The interview data were analysed using thematic analysis.[15] Major and minor themes were identified through an iterative process involving multiple readings of the transcripts. These themes were compared between and within the data,[15] and agreed by both authors.

Illustrative quotations were selected to support the data interpretation which were sent to the respective participants to check that they could not be identified by the content. Participants were allocated numbers (P1–P13) to preserve anonymity.

### Patient and public involvement
There was no patient or public involvement in this study.

## RESULTS
The 13 participants included 10 associate professors and three professors from a range of specialties. Most (seven) were aged between 40 and 50; two were aged between 30 and 40; and four were aged 50+. In terms of ethnicity, participants described themselves as White (1), White British (8), White Other (2) or Asian (2). The majority of participants (7/13) had been in their current post between 1 and 5 years; two for 1 year or less, and four participants had been in post for 5 or more years. No further details are given due to the rarity of women in these roles in some specialties which would risk making them identifiable.

Analysis of the data confirmed previous research suggesting that barriers and facilitators to progress are experienced at individual, interpersonal, institutional/procedural and societal levels.[6] These are discussed in more detail below.

### Barriers at an individual level
#### Juggling roles
Although the original aim was to interview female medical clinical associate/professors with caring responsibilities, some participants claimed not to have any because they did not have children. However, it emerged through the interviews that almost all the participants cared for either children, parents or other adults, or a combination. While participants agreed that they enjoyed the mix of clinical and academic work which meant that they were 'never, ever bored' (P12), even without caring

responsibilities, their role was described as 'trying to serve two masters…it's extremely difficult' (P6). Participants reported difficulties in making it clear what their role as a clinical academic entailed, which sometimes led to conflicts: 'people expect the same as they would of a full-time person in each of these roles' (P10).

The addition of 'the third role, at home, that everyone knows about, but nobody talks about' which was described as 'near enough full-time in itself' (P10) brought added challenges. Participants who shared parenting with their male partners described themselves as 'the default parent; the dropper-offer/picker-upper, and sorter-outer' (P10), bearing the brunt of childcare as well as managing the household.

Participants described additional caring responsibilities experienced as a result of the COVID-19 pandemic. Although children of key workers were allowed to attend nurseries and schools during lockdowns, one participant described how she needed to work from home when her child's class had to isolate after one person tested positive for COVID-19. Others described supporting clinically vulnerable family members and doing shopping and other tasks for elderly relatives to enable them to shield during the pandemic.

Issues around juggling these roles were further compounded by societal expectations that women should put childcare before their career, with some participants reporting that they were made to feel guilty by other people. The naturalisation of gendered roles was revealed through frequent references to their male partners being praised for doing cooking and childcare which were apparently perceived as 'women's work', even when both partners had full-time jobs.

### Lack of self-confidence

Some participants reported difficulties in building sufficient confidence to apply for an associate professor role and, even when successful, they suffered from 'imposter syndrome'. Relatedly, there were multiple mentions of 'luck', suggesting that some participants attribute their success to chance, rather than recognising their hard work and dedication as described by other participants in the study. It was suggested that this may be due to the way that girls are socialised to behave: 'boys are praised for things that we aren't praised for' (P12).

### Barriers at an interpersonal level
#### Discouragement

There was a recurring theme of participants being overlooked for promotion or discouraged from applying for senior posts. Some participants described conversations with both male and female colleagues where doubts were expressed about women's abilities to 'have it all'. For example, being advised that the clinical academic role was 'unsuitable for a woman with children' (P1) and 'I shouldn't apply because I would be too busy with my child' (P2). This view was echoed by a participant in this study who suggested that women needed to choose between pursuing a clinical academic career and having a family.

There were frequent references to being undermined, patronised and having achievements minimised; there were also reports of bullying. Participants felt that their complaints were sometimes dismissed: 'well, you know what he's like!' (P10) and attributed to 'a difficulty you are going through' (P2), rather than institutional problems.

### Barriers at institutional/procedural levels
#### Shortage of suitable clinical academic posts

Often the only way to get a clinical academic job was to successfully apply for a fellowship: 'without it, there was no guarantee of a job locally' (P11). However, participants had to prepare applications in their own time, thus disadvantaging those with caring responsibilities.

Although acknowledging that a lack of job security is not gender specific, 'in the clinical academic world, you're constantly looking for the next role' (P2), it was considered particularly challenging for women who were unable to move to another university because their partners could not, or would not, relocate. Anxiety over future career prospects was suggested as a possible reason why many women leave the clinical academic career route. Participants felt that positive action was needed to avoid attrition of talented female medical clinical academics such as additional fellowships, specifically for women. Other suggestions included alternative career paths to allow women to (re)join the clinical academic pathway later in their careers and reintroducing an intermediate grade between associate professor and professor, which 'might be a useful steppingstone' (P7).

### Masculine culture

Participants became aware of the lack of women at senior clinical academic levels as they progressed in their careers, even in predominantly female clinical specialties. They described a competitive culture as: 'survival of the cruellest' where success depended on navigating 'blocks and barriers' (P2), rather than nurturing people. Within this environment, participants noted how men were encouraged to express their opinions, whereas for women: 'if you speak too much, you're aggressive, but if you're shy you're weak' (P8).

### Promotion processes

The masculine culture was also said to influence promotion practices. Although one senior clinical academic noted that promotion committees were more likely to include women since the implementation of Athena SWAN initiatives, recently promoted participants described facing all-male interview panels which were said to be 'daunting' (P5). In addition, interviews were described as 'designed to intimidate the candidate, which particularly disadvantages women' (P1). One participant believed that women often disadvantage themselves by crediting their team effort and avoiding self-promotion. Suggesting that: 'it's to do with how we're socialised…boys are taught to think

that they're wonderful and we aren't', she recommended that female candidates prepare their applications 'like a man', present themselves 'as confidently as possible' and 'prepare for the possibility that the interview panel may behave fairly aggressively' (P12).

Participants felt that promotion criteria were biased against clinical academics when competing against full-time academics, and people with caring responsibilities. Comparisons were made with clinical progression which, although it might be delayed due to caring responsibilities, was not reliant on meeting academic criteria. In this respect, the H index was particularly concerning for women who had taken maternity leave and/or worked part-time, resulting in fewer publications, successful grant applications, etc, when compared with male candidates. It was suggested that 'pro-rata comparisons between candidates' (P12) would make it fairer for women who had worked part-time during their career. It was also pointed out that some specialties did not attract as much research funding, or have so many high-ranking journals, as other specialties. Therefore, it would be fairer to include different criteria such as research impact, for example, benefits for patient care.

### Accessing family-friendly policies

Some participants with caring responsibilities reported feeling reluctant to access family-friendly policies. For example, one participant framed her request to work from home 1 day each week in terms of being more productive academically, rather than enabling her 'to do the school run and interact with other mums' (P4). It was felt that working part-time was accepted more readily in the clinical setting than in clinical academia where participants feared that working less than full time might adversely affect funding applications. Other participants felt that part-time working was not effective: 'the difficulty is that the work doesn't decrease, there's the same amount of work in fewer hours, for less pay' (P12). In addition, one participant recalled 'paying for full-time childcare but getting paid 50% of my salary' (P13) in case work meetings or conferences occurred on 'non-work' days.

### Facilitators at an individual level

When asked how they had overcome challenges, there were multiple mentions of personal qualities such as resilience: 'anybody else would've walked away, but my personality is more robust than that' (P2). Some participants attributed their sense of self-belief to their upbringing: 'It depends on what your parents instil in you, in terms of what you can achieve…they didn't put any limitations in my head' (P3). Supportive partners and wider family members, along with reliable childcare, were also considered fundamental to career progression.

### Facilitators at an interpersonal level

Participants reported that encouragement from other people, including colleagues and coaches, had helped them to build confidence. Good mentorship was described as 'absolute key if you want to succeed' (P5). Many had found their own mentors through various sources including other university departments or institutions. Formal mentorship schemes were described as particularly helpful. However, some participants experienced difficulties in finding a suitable mentor, suggesting that there should be more rigorous processes in place.

In turn, participants enjoyed their role as mentors: 'you can actually facilitate and advise people…they're more likely to recognise that it's acceptable to ask for help' (P4). However, they felt that mentoring should be adequately resourced: 'there needs to be benefits for the mentor and the mentee, but there's no mechanism for that' (P13).

### Facilitators at institutional/procedural levels
#### Flexibility

Participants valued the flexibility of the clinical academic role, specifically the academic aspects which could be fitted around caring responsibilities much more easily than their strict clinical schedules would allow: 'that flexibility has really been key for me' (P1). This was particularly appreciated during the COVID-19 pandemic where participants were able to work from home: 'I have a lot more flexibility and control over my time than friends who chose to be full-time NHS consultants' (P3). Remote working became commonplace and 'much more acceptable' (P10) during the COVID-19 pandemic, with meetings held online described as 'a revelation…I'm not having to travel and fight for a car parking space…I'm more productive' (P6). It was hoped that these practices would continue after pandemic.

#### Athena SWAN initiatives

Both the universities from which the sample was drawn had achieved Silver Athena SWAN status by introducing initiatives such as increasing the presence of women in interview panels. However, one participant felt 'irritated' that she was often asked to be part of an interview panel as 'the token woman', and 'not because they felt I could bring anything to the interview process' (P4).

Other measures included having dedicated car parking spaces for people arriving after 09:00: 'knowing you can arrive at 9.15 and you don't have to park offsite somewhere' was described as a 'practical thing that really helps' (P13). However, the parking provision was considered inadequate, '5 spaces in the medical school carpark doesn't really cut it' (P13). Also, it was sometimes abused, 'people park there before 9 o'clock and just sit in the car until they're allowed' (P2).

Another facilitator was the directive to schedule meetings in core times: 'this Athena Swan initiative of meetings only being between 10 and 3 is just so wonderful, but whether people stick to them is another matter. I always stick to them; I never send meeting requests outside those hours' (P11). Although the idea was to enable people with caring responsibilities to attend meetings, another participant said: 'People don't stick to it. They say, "well

4 o'clock is the only time I can make". Well, that's my school pick up time and I've got nobody else that can help, so I don't have a choice' (P2).

Despite most participants welcoming these family-friendly policies, some were critical about the amount of support that was directed at parents: 'sometimes we feel like if we don't have kids…they're not interested in [supporting] us' (P6).

### Leadership development courses

Although individual support was appreciated, it was felt insufficient to deal with a 'male-heavy' environment (P5), and that what was needed was 'a change of culture' (P2). In this respect, leadership development courses were described as valuable for realising personal strengths and 'break[ing] down internal barriers that women are socially conditioned with' (P13). Participants felt that developing alternative leadership styles based on nurturing talent would improve the culture and encourage more women to pursue the clinical academic pathway: 'the biggest difference can be made by more representation' (P13).

### Attainable role models

Participants in this study were able to identify female role models within their universities, although there were concerns that some women may make success seem unattainable. It was therefore felt important that female leaders share their stories which include their failures

and setbacks along the way, as well as their tips for overcoming challenges.

In turn, participants discussed using their position to demonstrate an alternative culture:

I think we need to role model to people that you will never get an email from me at 6.30pm because that's dinnertime. It makes everybody else think it's okay, whereas if only one person is doing it, you're the odd one out. But if that's the culture of what's healthy to have a happy life, then that culture spreads. (P11)

They were also keen to encourage other female medical clinical academics following behind them: 'Now that I've climbed the ladder, the most important thing I can do is hold it tight, so that other women can climb it more easily' (P13).

Participants offered advice for aspiring female medical clinical academics, as summarised in table 1.

## DISCUSSION

This study has explored the barriers and facilitators experienced by female medical clinical academics from the perspectives of associate professors and professors. The majority of participants had caring responsibilities for children and/or other adults, reflecting a wider UK society where 75% of women have primary responsibility for children, and 58% are carers for elderly and disabled relatives.[16] Some of the challenges in combining clinical

**Table 1** Advice

| Advice | Illustrative quotes |
|---|---|
| Plan ahead | Start preparing your consultant CV well in advance of applying… see where the gaps are, for example, citizenship; what committees can I join? what else can I do that I'll be judged on? (P11) <br> Factor in a long time to write your application. Get advice from lots of people including men because… It's a game and you're playing by men's rules. (P12) <br> It's about how you move forward once you have got that post to move to the next level. (P7) |
| Be realistic | See the advantages of clinical academic work as well as the challenges. (P1) <br> Understand the ups and downs in advance and that there will be both. (P4) |
| Seek advice | It's not a sign of weakness to ask for help. (P4) <br> Don't be afraid that you're asking something stupid. (P8) |
| Find support | Have options in case you need childcare at short notice. (P3) <br> Just because we've given birth to a child doesn't mean that we have to be attached to them 24/7. It's okay for them to be cared for by a nursery. (P13) <br> Have a good support structure with your NHS colleagues. (P5) |
| Time management | Don't be afraid to ask for things to be rearranged to suit your work-life balance. (P11) <br> Have a clear plan to protect your academic time. (P5) |
| Persevere | Nothing is going to change overnight; it all takes time. (P4) |
| Ignore negativity | Don't see gender as a barrier. (P6) <br> Never listen to people who say work/life balance isn't possible; if you want it, you'll be able to do it. We read that, 'women can't have a family and a career and a life.' Well men have had that for centuries! (P13) |
| Encouragement | Anything is possible. You don't know what you're capable of—just try it. (P6) |
| Remember what is most important | Be attentive to your children, that's really important. When you look back on your life you won't necessarily think 'thank goodness I stayed late and did all those emails!'—you look back on things that really matter. Just keep thinking about what's important to you. (P10) |

and academic roles, such as competing pressures from academic and NHS employers, resonate with previous studies of clinical academic trainees, and were not gender specific.[17 18] However, additional challenges were experienced by participants who shouldered the majority of caring responsibilities, a situation that was exacerbated by the COVID-19 pandemic. Participants described having to balance the competing demands of work and home, including the 'mental load' of 'managing' household tasks.[19]

Interventions at institutional levels, including Athena SWAN initiatives such as core meeting times, were appreciated.[12] However, participants often felt that the onus was on them to insist on their implementation, thus reiterating previous studies.[20] This 'disjuncture between formal expectations and informal interactions' has been perceived as simply 'paying lip service' to gender equity, leading to calls for policies and strategies to be implemented across multiple levels of organisations[13] (pp 473, 475). Furthermore, findings echoed previous research suggesting that women are sometimes reluctant to access 'family-friendly' policies for fear of being perceived as conforming to gender stereotypes and perhaps as less committed to their roles than their male counterparts.[6] This study therefore supports recommendations for institutional-level interventions to address the perceived stigma of accessing services designed to support family-friendly working,[19] and to encourage men and women seeking a better work/life balance to use them.[7]

Although many participants had children, some did not. Data revealed frustration that efforts to support women to progress seemed to assume that all female clinical academics were mothers, suggesting a narrow view of issues that women face. This resonates with Roy's argument that motherhood is often presented as the problem for female academics, whereas the barriers preventing progression are generally located within the systems in which academics are trying to work.[13 21] This study's findings support this argument, confirming previous research indicating barriers such as a lack of suitable academic posts available locally, and inequitable promotion processes for women who take parental leave and/or work part-time.[6] It reveals a need for interventions at individual, institutional and national levels,[4 13] including altering 'structures modelled on the productivity of the white male scholar'[21] (p 29). Findings reinforce calls for transparency in the career process to allow forward planning, acceptance of pro-rata outputs by part-time workers, consideration of alternative promotion criteria and more flexibility in the clinical academic pathway to improve retention.[4 6]

Analysis of the data confirmed previous research suggesting that factors impacting on clinical academic progression are often influenced by wider issues.[6] For example, although a lack of self-confidence and 'imposter syndrome' were experienced at an individual level, contributory factors included interpersonal issues such as being excluded, discouraged and being overlooked

for promotion.[11 13] Societal influences were also identified, with women being more inclined to downplay their achievements,[4] and attributing their success to 'luck' rather than their hard work,[22] perhaps because of the way they are socialised to behave.[23] Findings support recommendations that women have 'role models to inspire, mentors to guide and advise, and sponsors to offer opportunities, even when women feel underqualified'[23] (p 404). The importance of finding the right mentor[11] was confirmed in this study, reinforcing calls for mentorship opportunities to be offered equitably at institutional level rather than relying on informal networking.[6] In addition, the value of mentoring should be formally recognised and appropriately resourced.

Participants described how they managed challenges in working in a male-dominant environment by adjusting their behaviour, downplaying their femininity and avoiding being perceived as 'too aggressive or too weak'[11] (p 1153). However, research suggests that organisations which foster 'dog eat dog' attitudes and masculine norms such as 'put work first' are associated with 'low cooperation, abusive leaders [and] work-family conflicts'[24] (p 515), whereas 'women leaders have a positive effect on organisational performance'[11] (p 1155); consequently, it is in everyone's interest to challenge the gender inequity at organisational levels. According to Alwazzan and Al-Angari, 'culture and leadership are two sides of the same coin'[10] (p 14); therefore, it follows that culture change is best achieved by having more female leaders. This study's findings also confirm the importance of role models,[22] to show that it is possible with careful planning and hard work to have a rewarding career with potential benefits for healthcare policy and patient care.[25] Female associate/professors in our study described developing and demonstrating alternative leadership styles and behaviours,[8 10] which may encourage other women to follow in their footsteps.

While societal issues around unequal distribution of domestic and caring responsibilities cannot be addressed by institutional-level changes, Ovseiko et al argue that encouraging all workers to take advantage of policies such as flexible and remote working may have a knock-on effect for equalising unpaid care work between male and female partners, bringing advantages for work–life balance for all employees, and not just women.[12] Research suggests that the COVID-19 pandemic has raised awareness of the challenges faced by parents and other carers, and that working from home has become more acceptable.[26] This study supports the recommendations that alternative ways of working such as online meetings and conferences should continue after pandemic,[26] thereby enabling greater participation by clinical academics, including those with caring responsibilities.

There are limitations to this study, most notably survivor bias, in that only women who had successfully negotiated the clinical academic ladder were interviewed.[8] Future research should seek the views of women who decided to leave the medical clinical academic pathway to find out

why they left, and what support might help them to return. The study was limited by the relatively small number of participants who were recruited from one geographical area. It would be helpful to repeat the study elsewhere, since the issue of gender inequality in clinical academia is prevalent (inter)nationally.[13] In addition, issues of intersectionality were not explored in this study.[27] However, the current lack of female associate professors/professors from ethnic minorities would necessitate extending the study nationwide to enable sufficient data to be gathered for a meaningful study.

This study was limited to female medical clinical academics. Therefore, future research should explore the experiences of female associate professors or professors who are nurses, midwives and allied health professionals.

## CONCLUSION

This study has revealed that although barriers to progression are experienced at an individual level, they are often influenced at broader institutional and societal levels. While societal attitudes are difficult and slow to change, their influence in workplace cultures can be challenged by supporting more women to progress into leadership positions where alternative ways of working and managing can be demonstrated. The facilitating factors identified in this study, along with advice from the lived experiences of female associate/professors, may go some way to addressing gender imbalances and supporting women to combine a medical clinical academic career with family life.

**Acknowledgements** The authors wish to thank all the women who participated in this study. We are also grateful to Dr Nicole Woitowich and Professor Christi Deaton for their helpful feedback on a previous version of this manuscript.

**Contributors** Both authors contributed to the study design and data analysis. DT collected the data and drafted the manuscript. ER contributed to comments and edits. Final approval was given by both authors. DT acted as guarantor.

**Funding** This work was supported by STEMM-CHANGE at the University of Nottingham (to DT), one of the 11 projects within the EPSRC Inclusion Matters portfolio, and the National Institute for Health Research (NIHR) Applied Research Collaboration East Midlands (ARC EM; to DT and ER) (grant number NIHR200171).

**Disclaimer** The views expressed are those of the authors and not necessarily those of the NIHR or the Department of Health and Social Care.

**Competing interests** None declared.

**Patient consent for publication** Not applicable.

**Ethics approval** This study involves human participants and was approved by the University of Nottingham Faculty of Medicine and Health Sciences Research Ethics Committee (reference FMHS 83-0820). Interviewees were informed about the purpose of the research, their right to withdraw from the study and how their (anonymised) data would be used; also that the data would be stored confidentially on secure university systems. Participants gave informed consent to participate in the study before taking part.

**Provenance and peer review** Not commissioned; externally peer reviewed.

**Data availability statement** Data are available upon reasonable request. Some anonymised data are available upon reasonable request.

**ORCID iD**
Diane Trusson http://orcid.org/0000-0002-6995-1192

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
