## [Reviewer comments · BMJ Open]

ARTICLE DETAILS

TITLE (PROVISIONAL)	A qualitative study exploring barriers and facilitators to progression for female medical clinical academics: Interviews with female Associate Professors and Professors.
AUTHORS	Trusson, Diane; Rowley, Emma

VERSION 1 – REVIEW

REVIEWER	Woitowich, Nicole Northwestern University Feinberg School of Medicine, Medical Social Sciences
REVIEW RETURNED	23-Oct-2021

GENERAL COMMENTS	Summary: In this manuscript, Trusson and Rowley conduct a qualitative study to examine factors which promote or hinder the advancement of women faculty in academic medicine in the UK and describe themes which span individual and institutional levels. The barriers identified in this study include caregiving responsibility, workplace culture, and male-biased promotion practices, while facilitators of advancement included mentorship, workplace flexibility, leadership skill development. The authors contextualize their findings across work which has explored gender equity within academic medicine and more broadly across male-dominated work environments. Evaluation: Overall, this manuscript is well-written and the methods, results, and discussion are presented clearly. Please find a few suggestions which may strengthen the manuscript below: 1) The authors indicate that “No further details [about the participants] are given due to the rarity of women in these roles in some specialties which would risk making them identifiable.” While I do not disagree with this rationale (and appreciate that the authors recognize the lack of intersectionality as a study limitation), I think it is important to acknowledge the overall ethnic/racial composition of the group.2) The authors mention in the methods section that the, “timing of the research coincided with the Covid-19 pandemic.” Could the authors provide the specific time frame in which the interviews took place?3) I think the discussion of Athena SWAN initiatives is an interesting component of this work (as it pertains to the implementation and utility of a policy specifically designed to foster gender equity within higher education). Within the discussion, the authors state that the participants, “often felt that the onus was on them to insist on their implementation.” I feel that this may be an overstatement, compared to what is presented within the results.
---

	Are there other key observations which should be incorporated into the results to bolster this claim?
--	---

REVIEWER	Deaton, Christi University of Cambridge, Cambridge Institute of Public Health, School of Clinical Medicine
REVIEW RETURNED	04-Nov-2021

GENERAL COMMENTS	This is a clearly written discussion of facilitators and barriers to clinical academic progression to senior roles (Assoc Prof/Prof), but a few weaknesses need to be addressed. The term 'clinical academic' is used (I think) only in the context of Medicine, although the term applies widely to nurses, AHPs, psychologists, pharmacists, etc. in similar roles. When it says female AP/P were contacted at 2 major universities, does this refer to those in medical schools? I think discussion and conclusion needs to state that this is the experience of female medical clinical academics. Although you state that 13 is a high response rate, it is impossible to judge without knowing how many were contacted by email. More detail on the participants could be provided without compromising their identities, e.g. how many were Profs, etc., range of ages and mean time in posts. Profs may also have come through in a pre-Athena Swan era, were there any differences in their comments from APs? The authors state that analysis was by Framework, but the description of multiple readings leading to identification of themes does not fit with the structured, multi-step process of Framework analysis (e.g. line by line coding). This sounds like a thematic analysis, which is valid if not as structured and auditable as Framework analysis. Very little is said about the clinical aspects, but on-call and busy services must also affect work-life balance. It would be nice to know if those with long experience think that the situation has improved, or if anything has changed from previous studies in early 2000s. Given that this was conducted during the pandemic, how did that potentially influence your work? Much has been written regarding disadvantages for women during the pandemic due to shouldering greater burden of caring for children and/or ageing or shielding relatives. I think you have to add the small sample size as a limitation. I think these changes will strengthen the paper, and although limited as noted, it is important to recognise that barriers are still perceived by women in medical clinical academic roles.
---

VERSION 1 – AUTHOR RESPONSE

Reviewer: 1

Dr. Nicole Weitowich, Northwestern University Feinberg School of Medicine

Comments to the Author:

Summary: In this manuscript, Trusson and Rowley conduct a qualitative study to examine factors which promote or hinder the advancement of women faculty in academic medicine in the UK and describe themes which span individual and institutional levels. The barriers identified in this study include caregiving responsibility, workplace culture, and male-biased promotion practices, while facilitators of advancement included mentorship, workplace flexibility, leadership skill development. The authors contextualize their findings across work which has explored gender equity within academic medicine and more broadly across male-dominated work environments.

Evaluation:

Overall, this manuscript is well-written and the methods, results, and discussion are presented clearly. Please find a few suggestions which may strengthen the manuscript below:

Response: Thank you for these positive comments.

1) The authors indicate that “No further details [about the participants] are given due to the rarity of women in these roles in some specialties which would risk making them identifiable.” While I do not disagree with this rationale (and appreciate that the authors recognize the lack of intersectionality as a study limitation), I think it is important to acknowledge the overall ethnic/racial composition of the group.

Response: Details of the overall ethnic/racial composition of the sample have been added.

2) The authors mention in the methods section that the, “timing of the research coincided with the Covid-19 pandemic.” Could the authors provide the specific time frame in which the interviews took place?

Response: The interviews were carried out between October 2020 and February 2021. This information has been added to the manuscript.

3) I think the discussion of Athena SWAN initiatives is an interesting component of this work (as it pertains to the implementation and utility of a policy specifically designed to foster gender equity within higher education). Within the discussion, the authors state that the participants, “often felt that the onus was on them to insist on their implementation.” I feel that this may be an overstatement, compared to what is presented within the results. Are there other key observations which should be incorporated into the results to bolster this claim?

Response: Thank you for this observation. Data extracts which illustrate this argument have been added to the results section to substantiate this claim.

Reviewer: 2

Prof. Christi Deaton, University of Cambridge

Comments to the Author:

This is a clearly written discussion of facilitators and barriers to clinical academic progression to senior roles (Assoc Prof/Prof), but a few weaknesses need to be addressed. The term 'clinical academic' is used (I think) only in the context of Medicine, although the term applies widely to nurses, AHPs, psychologists, pharmacists, etc. in similar roles. When it says female AP/P were contacted at 2 major universities, does this refer to those in medical schools? I think discussion and conclusion needs to state that this is the experience of female medical clinical academics.

Response: Thank you for this positive feedback and the observation that further clarification was needed about the study sample. The manuscript has been amended to make it clear that the participants were all female medical clinical academics. However, picking up your point, the exploration of experiences of female Associate/Professors who are nurses, midwives or AHPs has been flagged up as an area for future research.

Although you state that 13 is a high response rate, it is impossible to judge without knowing how many were contacted by email. More detail on the participants could be provided without compromising their identities, e.g. how many were Profs, etc., range of ages and mean time in posts.

Response: The emails were distributed by administrators in the respective Schools of Medicine and went to all Professors and Associate Professors rather than making assumptions about whether they identified as female. However, only respondents who met the criteria were asked to participate in interviews. This explanation has been added to the manuscript. Demographic information has also been provided.

Profs may also have come through in a pre-Athena Swan era, were there any differences in their comments from APs?

Response: Care needed to be taken to avoid identifying the 2 Professors who were in post prior to 2011. However, the changes they had observed since the introduction of Athena SWAN initiatives have been incorporated into the analysis. Overall, similar issues were raised in interviews, regardless of the participant's time in post.

The authors state that analysis was by Framework, but the description of multiple readings leading to identification of themes does not fit with the structured, multi-step process of Framework analysis (e.g. line by line coding). This sounds like a thematic analysis, which is valid if not as structured and auditable as Framework analysis.

Response: References to Framework analysis have been removed and replaced by thematic analysis.

Very little is said about the clinical aspects, but on-call and busy services must also affect work-life balance.

Response: The data have been re-examined and appropriate comments have been added that address this point. Specifically, we highlight that part-time working patterns are considered more acceptable in clinical settings but also how participants valued the flexibility afforded by the academic aspects of their roles in contrast to colleagues with strict clinical schedules, particularly during the Covid-19 pandemic.

It would be nice to know if those with long experience think that the situation has improved, or if anything has changed from previous studies in early 2000s.

Response: Two professors had been in post for over 10 years, but it has been necessary to avoid identifying participants, given the small sample. Their transcripts have been re-examined and a comment added around having women on promotion committees. Otherwise, there were no specific comments regarding changes over the past two decades.

Given that this was conducted during the pandemic, how did that potentially influence your work? Much has been written regarding disadvantages for women during the pandemic due to shouldering greater burden of caring for children and/or ageing or shielding relatives.

Response: Thank you for this observation. The methods section has been amended to show how the pandemic affected recruitment and data collection.

The data have been re-examined and additional details have been provided in the results section which reveal participants' additional caring responsibilities experienced during the pandemic.

I think you have to add the small sample size as a limitation.

Response: This limitation has been added.

I think these changes will strengthen the paper, and although limited as noted, it is important to recognise that barriers are still perceived by women in medical clinical academic roles.

Response: Thank you for this valuable feedback. We agree that the paper is much stronger as a result of incorporating both reviewers' suggestions for improvement.

VERSION 2 – REVIEW

REVIEWER	Deaton, Christi University of Cambridge, Cambridge Institute of Public Health, School of Clinical Medicine
REVIEW RETURNED	05-Feb-2022
GENERAL COMMENTS	Thank you, the authors have answered the reviewers' questions and I recommend acceptance.